# Automated Segmentation of Microvessels in Intravascular OCT Images Using Deep Learning

**DOI:** 10.3390/bioengineering9110648

**Published:** 2022-11-03

**Authors:** Juhwan Lee, Justin N. Kim, Lia Gomez-Perez, Yazan Gharaibeh, Issam Motairek, Gabriel T. R. Pereira, Vladislav N. Zimin, Luis A. P. Dallan, Ammar Hoori, Sadeer Al-Kindi, Giulio Guagliumi, Hiram G. Bezerra, David L. Wilson

**Affiliations:** 1Department of Biomedical Engineering, Case Western Reserve University, Cleveland, OH 44106, USA; 2Department of Biomedical Engineering, The Ohio State University, Columbus, OH 43210, USA; 3Department of Biomedical Engineering, Faculty of Engineering, The Hashemite University, Zarqa 13133, Jordan; 4Cardiovascular Imaging Core Laboratory, Harrington Heart and Vascular Institute, University Hospitals Cleveland Medical Center, Cleveland, OH 44106, USA; 5Cardiovascular Department, Galeazzi San’Ambrogio Hospital, Innovation District Milan, 20157 Milan, Italy; 6Interventional Cardiology Center, Heart and Vascular Institute, University of South Florida, Tampa, FL 33606, USA; 7Department of Radiology, Case Western Reserve University, Cleveland, OH 44106, USA

**Keywords:** optical coherence tomography, microvessel, deep learning, segmentation, classification

## Abstract

Microvessels in vascular plaque are associated with plaque progression and are found in plaque rupture and intra-plaque hemorrhage. To analyze this characteristic of vulnerability, we developed an automated deep learning method for detecting microvessels in intravascular optical coherence tomography (IVOCT) images. A total of 8403 IVOCT image frames from 85 lesions and 37 normal segments were analyzed. Manual annotation was performed using a dedicated software (OCTOPUS) previously developed by our group. Data augmentation in the polar (*r*,*θ*) domain was applied to raw IVOCT images to ensure that microvessels appear at all possible angles. Pre-processing methods included guidewire/shadow detection, lumen segmentation, pixel shifting, and noise reduction. DeepLab v3+ was used to segment microvessel candidates. A bounding box on each candidate was classified as either microvessel or non-microvessel using a shallow convolutional neural network. For better classification, we used data augmentation (i.e., angle rotation) on bounding boxes with a microvessel during network training. Data augmentation and pre-processing steps improved microvessel segmentation performance significantly, yielding a method with Dice of 0.71 ± 0.10 and pixel-wise sensitivity/specificity of 87.7 ± 6.6%/99.8 ± 0.1%. The network for classifying microvessels from candidates performed exceptionally well, with sensitivity of 99.5 ± 0.3%, specificity of 98.8 ± 1.0%, and accuracy of 99.1 ± 0.5%. The classification step eliminated the majority of residual false positives and the Dice coefficient increased from 0.71 to 0.73. In addition, our method produced 698 image frames with microvessels present, compared with 730 from manual analysis, representing a 4.4% difference. When compared with the manual method, the automated method improved microvessel continuity, implying improved segmentation performance. The method will be useful for research purposes as well as potential future treatment planning.

## 1. Introduction

Microvessels in the plaque area have been linked plaque rupture and intraplaque hemorrhage and are thought to be an indicator of plaque progression. Microvessels promote plaque formation by supplying red blood cells, the membranes of which serve as a rich source of free cholesterol [1]. In a pathohistological study, Sluimer et al. found that microvessel density was associated with coronary plaque progression [2]. In addition, they reported that the compromised structural integrity of the microvascular endothelium may explain the microvascular leakage responsible for rapid plaque progression in advanced coronary atherosclerosis by intraplaque hemorrhage [2].

A microvessel is a signal-free tubuloluminal structure with no connection to the vessel lumen that is visible on more than three continuous cross-sectional intravascular optical coherence tomography (IVOCT) frames [3]. Thanks to its near histological resolution (axial: 10 µm, lateral: 20–40 µm) and optical contrast [4], IVOCT is the only imaging modality that allows a unique assessment of microscopic structures (e.g., thin cap fibroatheroma (TCFA), macrophage, and microvessel). Therefore, IVOCT can provide an unprecedented opportunity to assess this feature of plaque vulnerability.

Although IVOCT allows for the identification of microvessels, it is quite challenging to identify them for research or clinical purposes. First, a single IVOCT pullback contains 300–500 image frames. Depending on experience, a clinician typically needs more than 20 min to review a pullback and carefully label microvessels. This timeframe is impossible to meet during a clinical procedure and extremely time-consuming for clinical research, especially with a large number of patients. Second, manual analysis of microvessels can be subject to inter-and intra-observer variability. This degree of variability creates a confound for widespread quantitative and visual evaluation, especially considering the variable experience and threshold for detection among cardiologists. The quantitative evaluation of microvessels can help various clinical research studies to elucidate the underlying mechanisms and factors that cause plaque progression. When compared with manual analysis results, automated, consistent measurements will improve the power of such studies to examine differences.

The goal of this research is to create an automated method for detecting microvessels in IVOCT images. Our approach includes pre-processing, data augmentation, deep learning identification of microvessel candidates using a carefully annotated dataset, and classification of microvessel candidates. For the first time, we will be able to automatically detect microvessels across an entire lesion, which will provide an opportunity to perform a more comprehensive assessment of coronary plaque progression.

## 2. Materials and Methods

### 2.1. Study Population and Manual Labeling

The images are from the TRiple Assessment of Neointima Stent FOrmation to Reabsorbable polyMer with Optical Coherence Tomography (TRANSFORM-OCT) trial [5]. The study included 90 patients who had undergone IVOCT examination and had stable angina with documented ischemia or acute coronary syndromes. The presence of unprotected left main disease, chronic total occlusion, baseline serum creatinine >2.0 mg/dL, life expectancy <18 months, and unsuitability for OCT imaging (at the investigator’s discretion) were major exclusion criteria. The final analysis of 79 patients included 8403 image frames from 85 lesions and 37 normal segments. IVOCT images were obtained using a frequency-domain OCT system (ILUMIEN OPTIS, Abbott Vascular, Santa Clara, CA, USA), which uses a tunable light source that ranges from 1250 to 1360 nm. The imaging pullback was performed at a frame rate of 180 fps, pullback speed of 36 mm/s, and axial resolution of about 20 µm. This study was conducted in accordance with the Helsinki Declaration and with the approval of the University Hospitals Cleveland Medical Center’s Institutional Review Board. Written informed consent was waived from all patients.

Manual annotation was carried out using Optical Coherence Tomography Plaque and Stent (OCTOPUS) software previously developed by our group [6]. An expert cardiologist (10+ years of experience) manually labeled each image from the Cardiovascular Imaging Core Laboratory, University Hospitals Cleveland Medical Center, a leading IVOCT analysis laboratory with over 3000 clinical trial cases analyzed. To evaluate the intra-observer agreement of manual annotation, the cardiologist repeated manual assessments on each series of IVOCT images at a three-month interval. The consensus document [5] defined a microvessel as a signal-free tubuloluminal structure without a connecting lumen that was visible on more than three continuous IVOCT image frames. Pixels in all other regions that did not meet these criteria were assigned an additional class of “other”, allowing for the creation of a binary semantic segmentation.

### 2.2. Data Augmentation

Our previous study found that data augmentation significantly improved deep learning segmentation performance in IVOCT images [7]. In this study, data augmentation was first used on raw IVOCT image data in the polar (*r*,*θ*) domain to ensure that lesions appear from all possible angles and to improve the spatial invariance of methods. To do this, all of the raw polar (*r*,*θ*) images were concatenated to form one large 2D array, where *r* represents tissue depth and the *θ* is catheter rotation, which rotates from 0 to *N* × 360°, where N is the number of input images. We then changed the offset angle to extract new polar image frames with no data loss or distortion. Specifically, we shifted the starting A-line nine times by 55 A-line increments. Training polar images were then extracted from the 2D array.

### 2.3. Pre-Processing

We applied pre-processing previously proposed by our group [8,9]. To effectively identify the appropriate tissue region of interest for processing, we used polar (*r*,*θ*) rather than anatomical (*x*,*y*) IVOCT images. The steps are now identified. (1) The guidewire and corresponding shadow region were detected using dynamic programming [10] and removed because they provide no useful information. Dynamic programming is an algorithmic technique for solving an optimization problem by breaking it into simpler sub-problems [11]. Briefly, the first boundary of the guidewire was detected and the small window mask was used to find the second boundary. (2) The lumen boundary was detected using the deep learning method previously described by our group [7]. (3) To align all rows with the same starting pixel along the radial direction, each A-line was pixel-shifted to the left. Pixel-shifting reduced the size of the region of interest (ROI), simplified subsequent image processing, and aligned tissues so that different lesions resembled the network more. (4) Because of the limited penetration depth of IVOCT, we chose a specific range (1.5 mm, 300 pixels) in the *r* direction as the ROI. (5) A Gaussian filter with a kernel size of (7,7) and a standard deviation of 1 pixel was used to reduce speckle noise. The pre-processing output was directly fed into the semantic segmentation deep learning model for determining microvessel candidates.

### 2.4. Segmentation of Microvessel Candidates Using DeepLab-v3+

As shown in Figure 1, we used DeepLab-v3+ semantic segmentation to identify microvessel candidates. The DeepLab-v3+ takes the encoder and decoder modules to extract features and refine the segmentation results. This model combines the advantages of multiple dilated convolutions and the sharp object boundaries achieved from the encoder-decoder setup. We briefly describe the three critical stages as follows. (i) Atrous convolution works with space inside the filter, which can have a wide field of view and can be modified adaptively by varying the rate value. This convolution allows the network to capture multi-scale contextual information while also generalizing standard convolution operations. (ii) The Atrous spatial pyramid pooling module combines the atrous spatial pyramid pooling module and image-level features at various sampling rates and an effective field of view. By incorporating multi-scale contextual information from the feature map, this module can improve the segmentation performance of object boundaries. (iii) Encoder–decoder. The encoder uses atrous convolution to extract the important information, such as the objects in the image and their locations at various resolutions. The decoder creates an output label having the same size as the input image (Figure 1). Here, we used the Xception [12] as the primary feature extractor (backbone network), which replaces conventional inception modules with separable depth convolutions. Batch normalization and rectified linear unit (ReLU) were performed after each convolution. Because it provides context for a decision, the receptive field is an important factor in determining segmentation performance. In this study, the receptive field of our network covered the entire pre-processed 2D image.

### 2.5. Classification of Microvessel Candidates Using a Shallow CNN

After identifying microvessel candidates, we used a simple convolutional neural network (CNN) model developed by our group to classify a 2D bounding box on each candidate as either microvessel or non-microvessel [13]. We created a rectangular bounding box for each microvessel candidate based on its size in (*r*,*θ*) space. The candidate was always in the center of the bounding box. Each bounding box became an input to the CNN classifier after subsequent resizing for training (Figure 2). The CNN classifier used in this study had three convolutional layers, two maximum pooling layers, and one fully connected layer. The input size of the network was set to (30 × 30 × 1) and the output was binary classification (1: microvessel and 0: non-microvessel). We padded all of the edges of a bounding box by 3 pixels before processing. The padding values were all obtained from the pre-processed input image. Convolutional layers in our model had the same filter size (3 × 3), but a different number of filters (8, 16, and 32), and we extracted the relevant feature maps with a stride of 2 pixels. After a convolutional operation, batch normalization and ReLU layers were added to accelerate training and reduce the sensitivity to network initialization [14]. To reduce dimensionality, the maximum pooling layer with a pool size of 2 pixels was later implemented. This layer prevented overfitting during training and kept the network insensitive to small transformations, distortions, and translations [14]. The fully connected layer was followed by the final convolutional layer. For the last layer, softmax activation and two output units were used.

To improve classification performance, we also used data augmentation on bounding boxes containing microvessels during network training (Figure 3). As a microvessel segment can be in any orientation, each bounding box was rotated six times from 30° to 180° with a 30° interval, giving seven times the number of positive cases. Using this approach, the number of bounding boxes for training between microvessel and non-microvessel increased from 1:7 to 1:1. Figure 3 depicts typical procedures of data augmentation for bounding boxes with microvessels and candidate classification.

### 2.6. Network Training

The adaptive moment estimation (ADAM) optimizer was used to optimize the segmentation and classification networks [15]. The ADAM optimizer is a stochastic gradient descent optimizer that only requires first-order gradients with little memory. This method computes individual adaptive learning rates for various parameters and is applicable to a wide variety of non-convex optimization problems [15]. The initial learning rate, drop factor, and drop period were empirically chosen to be 0.001, 0.2, and 0.5, respectively, and the learning rate was gradually reduced. Every drop period, the initial learning rate was multiplied by a drop factor. The maximum number of epochs was set to 50. For the segmentation network, the class weights were computed as the inversed median frequency of class proportions, allowing larger class data to have a lower weight and smaller class data to have a higher weight in the loss function. To avoid overfitting in deep learning, we added a regularization term for the weights to the loss function. We set the training to end when the validation loss did not improve by more than 0.01% over five consecutive epochs or when a maximum number of epochs (50) was reached. In practice, the former stopping criterion was exercised. All image processing and network training steps were carried out using MATLAB (R2019b, MathWorks Inc.) on an NVIDIA GeForce TITAN RTX GPU (24 GB memory).

### 2.7. Performance Evaluation

To assess segmentation/classification performance and variation across samples, we used five-fold cross-validation. All segments (122) were divided into five independent subsets of 23–25 segments each, with training (70%), validation (15%), and testing (15%) assigned. Because images from the same segment can leave very similar images in the training, validation, and testing sets, subsets were based on segments rather than images. Each subset was held out for testing, while the remainder were used for training and validation. As a result, we precisely assigned each subset to the test set. This procedure was repeated five times to ensure that all pullbacks were used as training, validation, or test sets. We ensured that each segment was assigned to only one group.

The results were assessed in multiple ways. We first evaluated segmentation and classification automatically using pixel-wise classification sensitivity, specificity, accuracy, and segmentation with Dice, as defined below:(1)$$Sensitivity=\frac{TP}{TP+FN}$$
(2)$$Specificity=\frac{TN}{TN+FP}$$
(3)$$Accuracy=\frac{TP+TN}{TP+TN+FP+FN}$$
(4)$$Dice=\frac{2TP}{2TP+FP+FN}$$

Above, *TP* is the number of true positive pixels, *TN* is the number of true negatives, *FP* is the number of false positives, and *FN* is the number of false negatives. We reported the mean and standard deviation of all metrics over the five folds. In addition, to analyze the reproducibility of annotations, we measured the microvessel area within a single expert cardiologist to assess intra-observer variability. For this purpose, an expert cardiologist manually annotated microvessels across the whole of the IVOCT pullbacks, and manual assessment was repeated three months after the initial analysis. We then used linear regression analysis to compare the similarity of microvessel areas between the first and second assessments. We compared performance with and without pre-processing and data augmentation to examine the impact of image analysis steps. In addition, we compared segmentation results from a DeepLab-v3+ network to those from SegNet [16] and U-Net [17] to see if there was any performance variation when using different networks.

In addition to standard metrics, we carried out more clinically relevant assessments. We chose 60 plaque ROIs from all segments and compared the number of frames with microvessels present between manual and automated detections. Plaque ROI was defined as IVOCT plaque segments with lipidic or calcified plaques. Each plaque ROI had multiple frames without microvessels and only a few microvessel frames, for a total of 730 out of 2812 frames.

### 2.8. Statistical Analysis

The paired *t*-test was used to make group comparisons. A *p*-value of <0.05 was considered statistically significant. Statistical analysis was performed using R Studio (version 1.4.1717, R Foundation for Statistical Computing, Vienna, Austria). All data were presented as mean ± standard deviation. All authors had complete access to the results and accepted responsibility for their integrity and data analysis.

## 3. Results

We found a high intra-observer agreement within the same analyst on our manual annotation of a microvessel. The linear regression analysis showed an *R*^2^ of 0.909 (Figure 4). The mean bias of the microvessel area was only about 0.0003 ± 0.001 mm^2^ in a Bland–Altman analysis, and most measurements were included within the limits of agreement (Figure 4). The Dice coefficient was 0.970, indicating good agreement.

Pre-processing and data augmentation improved the segmentation results significantly. Figure 5 depicts the results of microvessel segmentation with and without data augmentation and pre-processing steps. Numerous false positives from the lumen, catheter, and guidewire shadow occurred in the absence of data augmentation and pre-processing (Figure 5B). The Dice coefficient was less than 0.1 (Table 1), implying that the raw polar IVOCT image is unsuitable for microvessel segmentation. Data augmentation on the raw polar image did not improve performance significantly (Figure 5C). Although pre-processing effectively removed the majority of false positives, microvessels were slightly overpredicted and some microvessels were missing (Figure 5D). With pre-processing, sensitivity decreased slightly from 92.1 ± 2.5% to 83.9 ± 7.7%, while the Dice coefficient increased from 0.07 ± 0.01 to 0.67 ± 0.17 (Table 1). When both data augmentation and pre-processing were used, the method showed significant improvements, with Dice of 0.71 ± 0.10, a sensitivity of 87.7 ± 6.6%, and a specificity of 99.8 ± 0.1% (Figure 5E and Table 1).

The candidate classification step helped to eliminate false positive microvessel blobs. Despite its simplicity, our CNN model performed admirably, with a sensitivity of 99.5 ± 0.3%, specificity of 98.8 ± 1.0%, and accuracy of 99.1 ± 0.5%, as shown in Table 2. Furthermore, our method produced only 1.2 ± 1.0% false positive candidates and 0.5 ± 0.3% false negative candidates. Figure 6 depicts the results of microvessel segmentation with and without the candidate classification step. The classification step eliminated the majority of residual false positives and the Dice coefficient increased from 0.71 to 0.73 (Table 1).

DeepLab v3+ outperformed other deep learning networks such as SegNet and U-Net (Figure 7). In this comparison, all networks were subjected to the same data augmentation and pre-processing steps, with the exception of the candidate classification step. The U-Net had the highest sensitivity of 91.1%, but generated many false positives (Figure 7B), resulting in a very low Dice coefficient (0.65, Table 3). Despite a few false positives across the images (Figure 7C), the SegNet performed similarly to the DeepLab v3+ with a Dice coefficient of 0.71 and a sensitivity of 88.3 (Table 3). Although the sensitivity was decreased to 85.5%, the DeepLab v3+ showed the highest Dice coefficient of 0.73 with only a few false positives. The saliency maps of three semantic segmentation networks are shown in Figure 8.

Our method demonstrated reasonable agreement with a full manual analysis for clinically meaningful metrics. We examined the results frame-by-frame for a subset of 60 plaque ROIs with and without microvessels. Our method produced 698 frames with microvessels out of 2812 frames compared with 730 from a manual analysis, giving a percent difference of only 4.4% (4.3% of false positives and 0.1% of false negatives). However, as argued below, we believe that some “false positives” were, in fact, true microvessel indications. In terms of agreement, there was no significant difference between lipidic and calcified plaque ROIs.

Figure 9 depicts three-dimensional visualizations of microvessels in a plaque using both manual and automated segmentation. As noted in the figure legend, the automated method gave a more continuous microvessel than the manual method, implying that our automated method outperforms manual annotations. For this reason, we suspect that several of the 4.3% “false positives” mentioned in the previous paragraph are not errors at all. Using the automated method on IVOCT images, it was possible to identify microvessels >7.4 mm in length. The diameter from IVOCT is estimated to be ~107.4 µm, but this number is overestimated owing to the resolution limit of IVOCT. Regardless, this analysis suggests that the plaque could be well perfused over its entire length.

## 4. Discussion

We have developed a promising method for detecting significant microvessels in coronary artery IVOCT images. Microvessels in plaque have been associated with plaque rupture and intraplaque hemorrhage and are thought to be indicative of plaque progression through the mechanisms described in the Introduction. The manual analysis of microvessels in IVOCT images is challenging, highlighting the importance of an automated image analysis method. Building on our previous studies of IVOCT image analysis [6,7,8,9,18,19,20,21,22,23,24,25], we developed an automated method using deep learning. The main findings of this study can be summarized as follows: (1) We found very high intra-observer agreement in our manual analysis, indicating that our labels are reasonably accurate. (2) Data augmentation and pre-processing aided in improving the segmentation results. (3) Candidate classification using a simple CNN assisted in removing residual false positives from candidate segmentation, resulting in improved overall performance. (4) The DeepLab v3+ network outperformed other deep learning methods like SegNet and U-Net. (5) When compared with manual assessment, our method had only 4.4% frame differences (4.3% of false positive frames and 0.1% of false negative frames), but several of the “false positive” frames are actually true positives because manual annotations are sometimes missed, as shown in the Results. Our method produced excellent results for microvessel segmentation while taking a reasonable amount of time to compute, implying that it could be a promising solution for both research and clinical applications.

Our two-step method produced very good agreement with manual analysis for microvessel segmentation. The quantitative results were very close to manual analyses, with a Dice coefficient of 0.73, sensitivity of 85.5%, specificity of 99.8%, and accuracy of 85.5%. We found that the raw polar IVOCT image was unsuitable for microvessel segmentation, with a Dice coefficient of 0.07. Most errors were caused by the lumen, catheter, and guidewire shadows, as shown in Figure 5, and data augmentation did not improve the results when applied to the raw IVOCT image. Pre-processing significantly enhanced the segmentation performance. Because of the limited penetration depth of the IVOCT signal, the pre-processing used in this study effectively removed regions (e.g., lumen, catheter, and guidewire) that can cause many false positives and limited the ROI to 1.5 mm (300 pixels) in the *r* direction without any data loss or distortion. As a result, pre-processing removed the majority of the segmentation errors observed when using the raw polar image. Data augmentation also improved the results significantly. Despite its simplicity, the candidate classification step proved useful in eliminating the remaining false positives. Remarkably, this step classified not only easily detectable regions such as lumen, catheter, and side branch, but also small calcifications with similar intensity characteristics to microvessels.

It is possible to determine multiple, long lengths of microvessels in plaque using our automated methods on IVOCT images (Figure 9). This observation, which has not previously been reported, suggests that these are active plaques that require significant blood perfusion. It will be interesting to see if these plaques are linked to future adverse findings in retrospective studies (e.g., plaque progression, plaque rupture, or in stent stenosis in the case of a treatment).

Our method should allow for more comprehensive coronary artery assessments than the promising reports produced thus far using manual analysis. In IVOCT images, microscopic plaques such as macrophage, TCFA, and microvessel are known to be among the strongest clinical indicators of plaque vulnerability [26]. According to reports, few isolated patterns are unlikely to be clinically significant, whereas a large accumulation within a thin fibrous cap may be of greater concern. Nakazato et al. [27] found a higher prevalence of adverse plaque characteristics when the macrophage coexists with the TCFA within a fibrous cap in their comparative study of IVOCT and computed tomography angiography. Kitabata et al. [3] found that patients with microvessels had more unstable angina (87% vs. 17%), a thinner fibrous cap (60 µm vs. 100 µm), and a higher risk of plaque rupture (50% vs. 28%). According to Uemura et al. [28], non-significant coronary plaques with angiographic progression had a significantly higher incidence of microvessel (76.9% vs. 14.3%, *p* < 0.01), TCFA (76.9% vs. 14.3%, *p* < 0.01), and macrophage images (61.5% vs. 14.3%, *p* < 0.01). TCFA and microvessel were highly correlated with subsequent luminal progression (*p* < 0.01) in univariate regression analysis [28]. Furthermore, Dong et al. [29] found that patients with high-grade rejection following cardiac allograft vasculopathy had significantly thicker intima (0.34 mm vs. 0.15 mm), a higher prevalence of macrophages (44% vs. 15%), and a higher prevalence of intimal microvessels (46% vs. 11%). Our previous studies proposed automated methods for quantifying macrophages and thin fibrous caps in IVOCT images [18,24,30]. As a result, combining quantitative analyses of macrophage, TCFA, and microvessel could provide an unprecedented opportunity to comprehensively assess coronary plaques in IVOCT images.

For clinical research applications, our automated method is computationally acceptable. On our computer system with a non-optimized code, automated microvessel analysis only takes about 0.16 s per frame (pre-processing, 0.05 s; candidate segmentation, 0.11 s; candidate classification, 0.003 s). Ideally, the proposed method can process an entire pullback of 375 frames in 60 s. However, processing lesions (segments) after specifying ROIs will be the preferred solution to reduce potential false positives from a whole pullback. The algorithm is simple to implement in IVOCT image analysis software (e.g., our OCTOPUS software [6]), where starting and ending frames for the analysis can be specified. Our method would be suitable for a wide range of clinical research projects, particularly those involving a large number of patients. Someday, algorithms like this might find their way into IVOCT treatment planning.

This study has some limitations. First, while this study used a large dataset, a future study using even more data with accurate annotation may improve the results even further. Second, for microvessel candidate segmentation, we used a standard 2D deep learning semantic segmentation (i.e., DeepLab v3+). Because it takes into account the spatial characteristics of microvessels, the results can be improved using more advanced 3D deep learning models (e.g., 3D U-Net).

## 5. Conclusions

We have developed an excellent microvessel segmentation method that can be easily applied to IVOCT pullbacks for a repeatable, fast analysis. The proposed method has the potential to enable highly automated and comprehensive plaque analysis in IVOCT images. We believe that this method will be useful for research applications and possibly for future treatment planning.

## Figures and Tables

**Figure 1 bioengineering-09-00648-f001:**
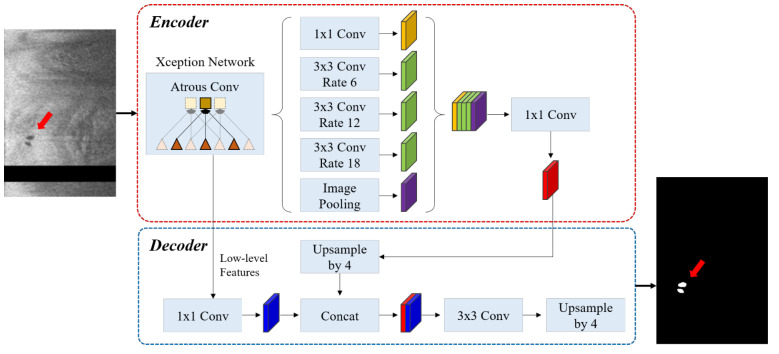
The DeepLab v3 plus network consists of atrous convolution, atrous spatial pyramid pooling, and encoder–decoder. The Xception network was used as the backbone network for feature extraction. The left and right figures are the pre-processed IVOCT image and predicted label, respectively (red arrow: microvessel). The sizes of input and output images are the same (200 × 448 pixels). In the input image, the black strip indicates the guidewire shadow removed during pre-processing.

**Figure 2 bioengineering-09-00648-f002:**
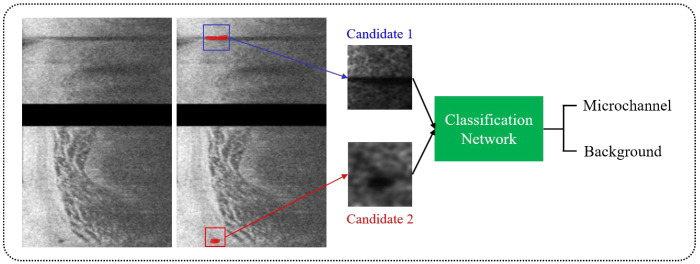
Classification of microvessel candidates using a simple CNN model. Each candidate was classified as either microvessel or non-microvessel using a CNN model (green) consisting of three convolutional, two maximum pooling, and one fully connected layers. The red box is a microvessel, and the blue box is a non-microvessel.

**Figure 3 bioengineering-09-00648-f003:**
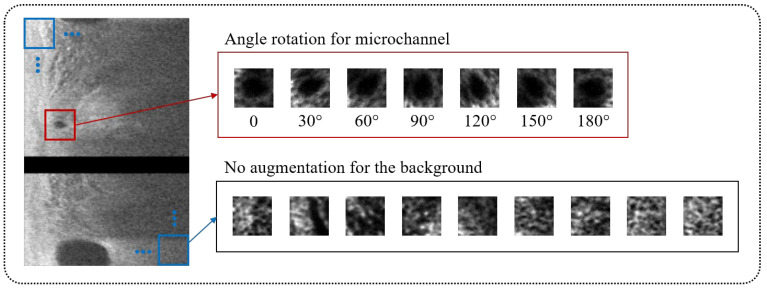
Data augmentation for candidate classification network. The bounding boxes with microvessel (red) were augmented by rotating its angle from 30° to 180° with a 30° interval, providing a seven times greater number of microvessel cases. The bounding boxes without microvessel (blue) were not augmented.

**Figure 4 bioengineering-09-00648-f004:**
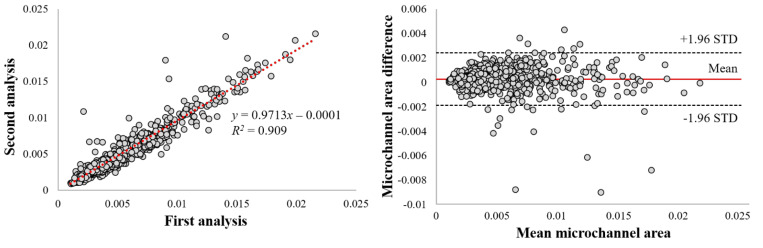
Intra−observer variability analysis of microvessel areas. Panels are as follows: (**left**) linear regression plot and (**right**) Bland−Altman plot. Linear regression gave *R*^2^ = 0.909. Dice was 0.970. In the Bland−Altman analysis and the mean bias was only about 0.0003 ± 0.001 mm^2^, about 4.7% of the mean area. This result suggests excellent manual reproducibility within a single analyst.

**Figure 5 bioengineering-09-00648-f005:**
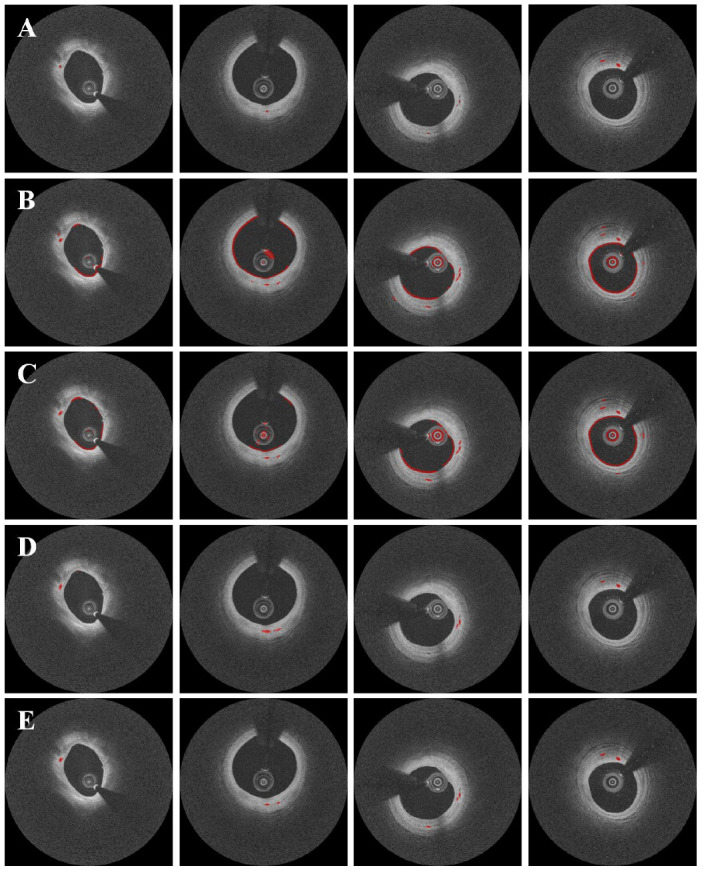
Segmentation results with and without data augmentation and pre-processing. Panels are results from (**A**) manual annotation, (**B**) raw polar IVOCT images, (**C**) data augmentation on the raw polar IVOCT image, (**D**) pre-processing alone, and (**E**) data augmentation and pre-processing. Both data augmentation and pre-processing improved the segmentation results. In some cases (second and third columns), our method detected microvessels that were missed during manual annotation, as confirmed by the analyst. Microvessels are labeled in red.

**Figure 6 bioengineering-09-00648-f006:**
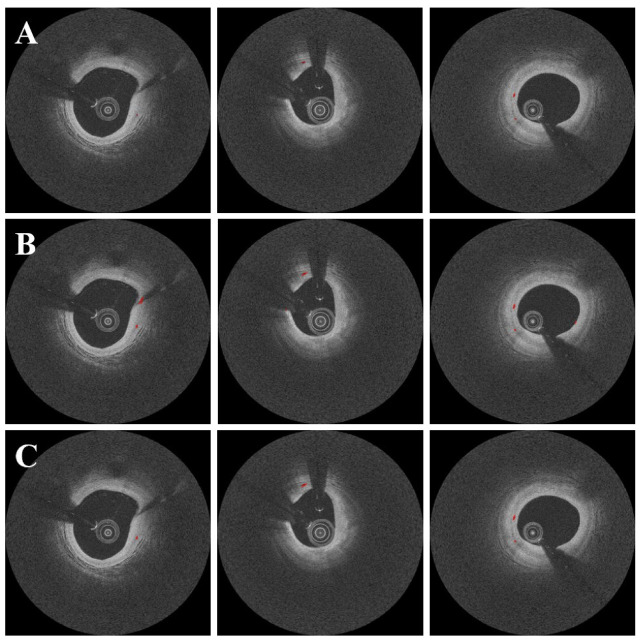
Segmentation results with and without the candidate classification step. Panels are as follows: (**A**) manual annotation, (**B**) results before candidate classification, and (**C**) results after candidate classification. There were some false positives from the candidate segmentation step (**B**), which were effectively ruled out with the candidate classification step (**C**). The red is a microvessel.

**Figure 7 bioengineering-09-00648-f007:**
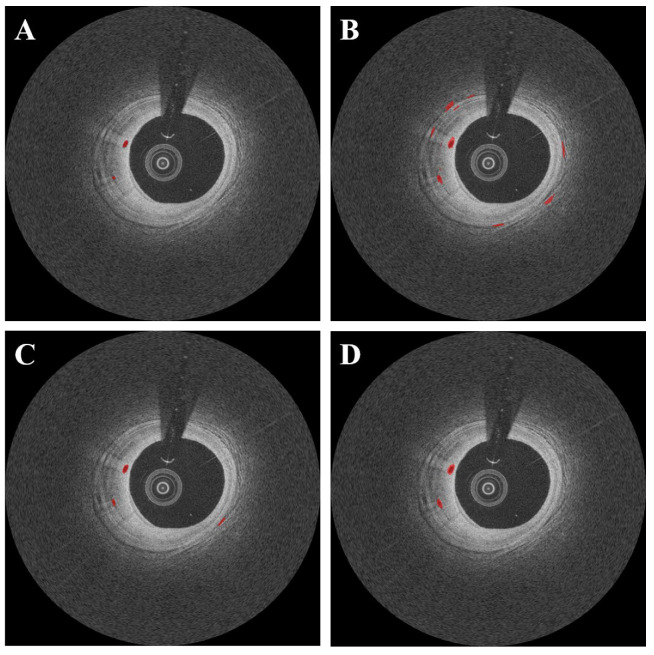
Segmentation results of microvessel according to different semantic segmentation models. Panels are as follows: (**A**) manual annotation, (**B**) results obtained using U-Net, (**C**) results obtained using SegNet, and (**D**) results obtained using DeepLab v3 plus. In (**C**), the SegNet had similar results as compared with DeepLab v3 plus; however, there were some misdetections. The U-Net (**B**) had the worst detection results among all deep learning models. The red is a microvessel.

**Figure 8 bioengineering-09-00648-f008:**
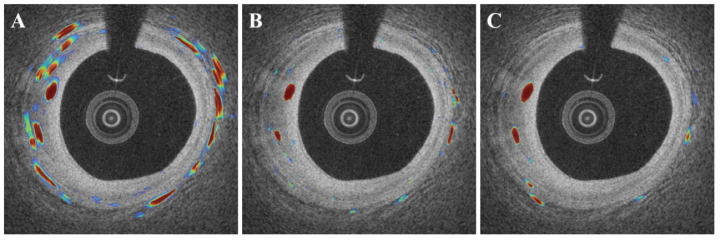
Examples of saliency maps obtained by different semantic segmentation networks ((**A**): U-Net, (**B**): SegNet, and (**C**): DeepLab v3+). The U-Net (**A**) showed a number of false activations along the angular rotation (*r*), while SegNet and DeepLab v3 plus focused on the specific regions with microvessels.

**Figure 9 bioengineering-09-00648-f009:**
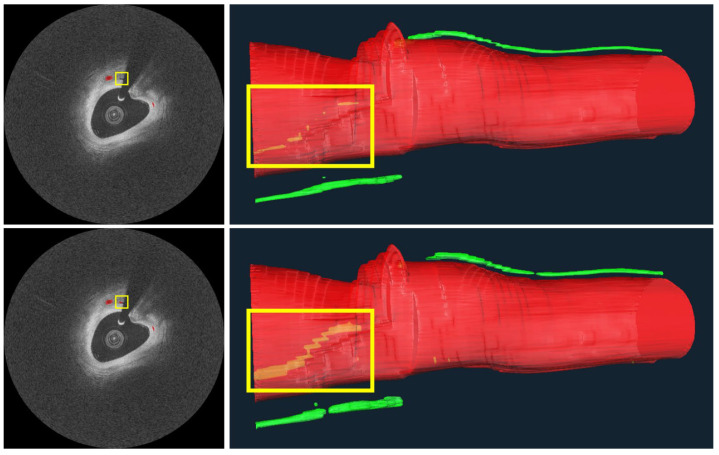
Microvessels from a representative IVOCT pullback in a plaque. On the right are 3D visualizations from manual (**top**) and automated (**bottom**) microvessel segmentations, as well as the vessel lumen. Microvessels look very similar between the top and bottom, except for within the yellow bounding boxes, showing a microvessel “behind” the vessel lumen. In this latter instance, the microvessel from the automated analysis is much more continuous than that from the manual analysis, indicating that the automated method detects more actual microvessel instances than the manual analysis. Missing annotations in the manual analysis are evident in the original IVOCT images on the left, where an annotation is missing despite image evidence. The longest microvessel is ~7.4 mm in length with a diameter of about ~107.4 μm. The red is lumen and the green is a microvessel.

**Table 1 bioengineering-09-00648-t001:** Performance metrics of pixel-wise microvessel segmentation over five folds, including the Dice coefficient, sensitivity, and specificity. The segmentation results were greatly improved using data augmentation and pre-processing. The candidate classification step removed some false positive segmentation spots, improving the pixel segmentation results.

Methods	Dice	Sensitivity (%)	Specificity (%)
Original image	0.07 ± 0.01	92.1 ± 2.5	96.2 ± 1.0
Data augmentation only	0.07 ± 0.03	88.9 ± 9.2	96.4 ± 1.8
Pre-processing only	0.67 ± 0.17	83.9 ± 7.7	99.7 ± 0.2
Data augmentation and pre-processing	0.71 ± 0.10	87.7 ± 6.6	99.8 ± 0.1
After candidate classification	0.73 ± 0.10	85.5 ± 6.9	99.8 ± 0.1

**Table 2 bioengineering-09-00648-t002:** Mean performance metrics of the candidate classification network over five folds, including the Dice coefficient, sensitivity, specificity, and accuracy. The network was trained/validated/tested on manually annotated data. Our CNN model showed remarkable classification results and was very useful to classify each candidate as either microvessel or non-microvessel. All metrics are reported as the mean and standard deviation over the five folds.

Method	Sensitivity (%)	Specificity (%)	Accuracy (%)
Candidate classification	99.5 ± 0.3	98.8 ± 1.0	99.1 ± 0.5

**Table 3 bioengineering-09-00648-t003:** Mean performance metrics of microvessel segmentation according to different semantic segmentation networks (U-Net, SegNet, and DeepLab v3 plus), including the Dice coefficient, sensitivity, and specificity. The DeepLab v3 plus model showed the highest Dice coefficient. All metrics were obtained after candidate classification.

Methods	Dice	Sensitivity (%)	Specificity (%)
U-Net	0.65 ± 0.19	91.1 ± 4.8	99.6 ± 0.3
SegNet	0.71 ± 0.11	88.3 ± 1.8	99.7 ± 0.1
DeepLab v3+	0.73 ± 0.10	85.5 ± 6.9	99.8 ± 0.1

## Data Availability

The data presented in this study are available upon request from the corresponding author.

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
