# Peer review of "Automated Segmentation of Microvessels in Intravascular OCT Images Using Deep Learning"

_bioengineering, 2022, doi:10.3390/bioengineering9110648_

Round 1

Reviewer 1 Report

The authors report automated segmentation of microvessels in IVOCT images using deep learning.

Figures 5 and 7 give the impression that the results of manual annotation and DeepLab v3 plus are very similar, but Figure 9 shows the difference.

In principle, it is a well-written paper, and the research is well-ordered. However, in the end, what is the rate at which the difference from manual annotation is produced as described above?

It says that the informed consent was obtained from all subjects, but how many subjects and where are their data presented?

Reviewer 2 Report

L125: please elaborate on the term “dynamic programming”

L239-254: F? T?

Table 1: is the Dice coefficient of 0.07 correct? Seems very low. Please check.

Table 3: What do you think to be the most effective picture segmentation metrics? The reader is confused by the high levels of sensitivity and specificity in Table 3. When discussing how DeepLab v3+ beats other deep learning networks (SegNet and U-Net) in the text, you should discuss all metrics used and explain the one you chose (i.e., Dice) to reach the above claim.

L334: what do you think is the reason for the increased performance of DeepLab v3+? Did the pre-processed images feed also U-Net and SegNet?

L341-347: please explain the “manual analysis” procedure. Was it performed by the same person? In Fig 4 it is written that this is the case (single analyst). Please provide this in the text beside the observer's level of experience.

General remark: Is the code free and accessible? If not, please consider it, as the effect of this work will be diminished otherwise.

Round 2

Reviewer 2 Report

Thank you for taking into account my comments. Congratulations for your work.